

# The motivation to inform others: a field experiment with wild chimpanzees

Derry Taylor[1,2], Sam Adue[2], Monday M'Botella[2], Denis Tatone[3], Marina Davila-Ross[4], Klaus Zuberbühler[1,2] and Guillaume Dezecache[2,5]

[1] University of Neuchatel, Neuchatel, Switzerland
[2] Budongo Conservation Field Station, Masindi, Uganda
[3] Central European University, Budapest, Hungary
[4] University of Portsmouth, Portsmouth, United Kingdom
[5] Université d'Auvergne (Clermont-Ferrand I), Clermont-Ferrand, France

## ABSTRACT

**Background:** Accumulating evidence indicates that some ape species produce more alarm behaviors to potential dangers when in the presence of uninformed conspecifics. However, since previous studies presented naturalistic stimuli, the influence of prior experience could not be controlled for.

**Method:** To examine this, we investigated whether apes (wild chimpanzees of the Budongo Forest, Uganda) would communicate differently about a novel danger (an unusually large spider) depending on whether they were with an uninformed conspecific. We tested nine adult males, four of which were exposed to the danger twice alone (Non-Social group), while the remaining five were exposed to the danger first alone and then in the presence of conspecifics (Social group).

**Results:** We found that both alarm calling and gaze marking (*i.e.*, persistent gaze after stimulus detection) were more persistent in the Social than Non-Social group, although the effect of condition only reached statistical significance for gaze marking, nonetheless suggesting that chimpanzees tailored their warning behavior to the presence of others, even if they were already familiar with the potential threat.

## INTRODUCTION

The ability to modify communicative signals conditional on the epistemic state of others is a highly adaptive trait for social animals. This is especially the case in situations of danger when knowledge states differ between members of a social group. For instance, animals sometimes communicate about the presence of danger to conspecifics, even if it poses no direct threat to the signalers themselves. In the wild, such a proclivity likely plays an important role for safely navigating environments, including those that are subject to novel anthropogenic disturbances, such as poaching or infrastructure developments (*e.g.*, *Hockings, Anderson & Matsuzawa, 2006*).

Converging evidence indicates that some chimpanzees and bonobos adjust their alarm behavior to the putative informational needs of others (*Crockford et al., 2012*; *Crockford, Wittig & Zuberbühler, 2017*; *Girard-Buttoz et al., 2020*), by communicating more frequently or elaborately when others appear to be unaware of the presence of a potential

Corresponding author
Derry Taylor,
derry.taylor@port.ac.uk

threat (a model snake) nearby. Relatedly, receivers expect alarm signals to be about something in their immediate surroundings, as evidenced by playback studies with different species (*e.g.*, *Seyfarth, Cheney & Marler, 1980*; *Crockford, Wittig & Zuberbühler, 2015*). However, while in field experiments conducted so far (*e.g.*, *Crockford et al., 2012*; *Girard-Buttoz et al., 2020*) factors such as number of experimental exposures to a predator stimuli and the presence or absence of startle responses were controlled for statistically, owing to the naturalistic nature of the stimuli individuals may have differed considerably in how much experience they had with the potential threat, opening the possibility that differences in previous experience with the danger may account for some of the variability in alarm responses. For example, more experienced individuals may call more when conspecifics move in the direction of a predator model because they have learned to associate this behavior with negative outcomes such as snake bites. Given chimpanzees live in a fission-fusion social system with demonstrated individual variation in territory use (*e.g.*, *Badihi et al., 2022*), we believe such individual differences in experience to be plausible, although the extent of this variation is of course unknown.

A number of field experimental studies have investigated this issue by placing models of natural predators on the predicted travel path of wild primates to study how the discoverer of the potential threat informs its naïve audience (*e.g.*, *Arnold, Pohlner & Zuberbühler, 2008*; *Quintero et al., 2022*). In one key study, *Crockford et al. (2012)* presented individual chimpanzees with a model of a dangerous snake, which led to patterns of warning behavior consistent with the callers' motivation to warn naïve group members of the imminent threat. Later, a similar result was obtained by *Girard-Buttoz et al. (2020)* who studied bonobos and chimpanzees (notably, a different population than was studied by *Crockford et al., 2012*) using a similar method, indicating this pattern is generalizable across chimpanzee populations and is also present in bonobos. In a similar study building on the work of *Crockford et al. (2012)* who presented stimuli only when conspecifics were present, *Schel et al. (2013)* presented wild chimpanzees with a dynamic (rather than static as was used by *Crockford et al. (2012)*) snake model when individuals were alone or in the company of others. The authors found there was no difference in alarm calling when the subject was alone compared to with others, suggesting alarm calling was not related to danger faced by others, and found instead alarm calling was largely influenced by the presence of others with whom they shared close social bonds. However, since prior experience cannot be controlled in experiments presenting naturalistic stimuli as explained above, it is difficult to ascertain the extent to which individuals communicate based on what others know (or do not know). For this, presentation of *novel* threats would provide valuable insights. Such an approach would enable clearer interpretations of experimental observations. For example, *Schel et al. (2013)* pointed out that uninformed individuals (*i.e.*, those who have not yet detected the snake) may have also been more likely to move in the direction of the snake, meaning the calling behavior of the 'detector' (*i.e.*, the individual who has detected the snake) may have been a response to imminent danger for others based on past experiences wherein uninformed conspecifics have had dangerous encounters with snakes after unknowingly moving in their direction, rather than their knowledge state *per se*. By manipulating experience, we can rule out such interpretations.

Several scholars suggested that communication in chimpanzees and other non-human animals should be interpreted as primarily driven by affective processes (*Goodall, 1986*; *Hammerschmidt & Fischer, 2008*). Under this reading, alarm behaviors are primarily produced as by-products of the heightened arousal state of the emitter. Contradicting this proposal, however, recent evidence shows that chimpanzees cease alarm behaviors such as calling and gaze marking (*i.e.*, persistent gaze towards the danger) when conspecifics are safely away from danger, which is at odds with this interpretation (*Crockford, Wittig & Zuberbühler, 2017*; *Schel et al., 2013*; *Townsend et al., 2017*; see *Girard-Buttoz et al., 2020* for similar results in bonobos). The debate is far from being currently settled, because obtaining objective measures of affective states, especially in communicative contexts, has remained an elusive pursuit (notwithstanding recent progresses in affective-state monitoring with free-ranging animals: *De Vevey et al., 2022*; *Dezecache et al., 2017*). A critical step in unveiling the proximate factors mediating alarm communication production is to experimentally isolate the alleged contribution of affect from the goal of informing others. We reasoned that, if there exists a genuine motivation in communicating valuable information to others, chimpanzees should produce communicative behaviors to uninformed others even after having been already exposed to the target referent, thus ruling out any surprise-mediated response. Differently put, provided that the stimulus is relevant to be communicated about to a potential audience, we should expect communication to occur in animals already aware of the target, and thus not immediately aroused by it, if their communicative behavior indeed reflects a motivation to inform others. In such a case, communicative behavior should be primarily affected not by the signaller's familiarity to the target stimulus, but by the knowledge state of the putative audience. While previous studies (*e.g.*, *Crockford et al., 2012*; *Girard-Buttoz et al., 2020*) controlled for surprise by including startle responses as a control variable in statistical models or in some cases conducting multiple trials in the same location, suggesting a knowledge-based motivation to inform beyond individual arousal, no studies have experimentally controlled for surprise, which would require multiple within-individual presentations of a novel stimuli.

To explore this, we designed a study in which $N = 9$ wild male chimpanzees of the Budongo Forest Reserve in Western Uganda were exposed to a novel animal model. This is similar to previous studies in the sense that we presented a naturalistic stimuli to wild chimpanzees, but different in the sense that it was an entirely novel animal not found in the territory of the study community, allowing us to control for variability in previous experience. First, all individuals were presented with the model alone. Then, all of them were then presented a second time with the spider. However, during their second exposure $N = 5$ subjects were in the company of naïve community members (Social group), whereas the remaining $N = 4$ subjects were tested alone for a second time (Non-Social group). If chimpanzee alarm behavior is mainly a direct response to external threats, then the second exposures should uniformly lead to weaker responses, regardless of the presence or absence of a naive audience (note that all of our 'audiences' are naïve given the novelty of the stimulus, meaning we cannot disentangle whether responses are due to the knowledge state of the audience or simply their presence). Instead, if chimpanzees are motivated to

inform others independent of their prior experience and regardless of their own inner state, we expected subjects in the Social group to show significantly more alarm behaviors compared to those in the Non-Social group, despite equal familiarity with the spider.

# METHODS

## Ethical note

Permission to conduct the study was granted by the Ugandan Wildlife Authority (Permit ID: COD/96/02) and the Uganda National Council for Science and Technology (Permit ID: NS599). The Sonso community has co-existed with humans for many decades. Its home range covers a large estate of an abandoned sawmill, the Budongo Conservation Field Station, and various settlements along the forest border. Individuals therefore regularly encounter human artifacts, some of which pose a grave threat, such as wire snares, mantraps and weapons used by farmers to drive away wildlife. All work complied with international standards such as the Association for the Study of Animal Behavior (ASAB) ethical guidelines and the UK Animal Scientific Procedures Act 1986, including taking all possible measures to follow the 3R tenets by using a design that required the fewest trials possible to answer the research question and limiting our study to a sub-sample of the overall population (*i.e.*, focusing only on male chimpanzees). We also followed taxon-specific guidelines, namely, the International Union for the Conservation of Nature's Best Practices for working with Great Apes.

## Study site

The study was carried out in the Budongo Forest Reserve, a moist semi-deciduous tropical forest in western Uganda, covering 428 km² at an altitude of 1,100 m, between 1°35′ and 1°55′ N and 31°08′ and 31°42′ E (*Eggeling, 1947*). Data were collected in the Sonso community between May 2015 and June 2015, and between April 2016 and May 2016. Habituation of this community to humans began in 1990, with the majority of individuals (approximate $N = 70$) well habituated to human observers (*Reynolds, 2005*).

## Individuals

Although more gregarious than females, males were more likely to be found travelling on their own, while females are typically accompanied with offspring (and other females when nulliparous). For this reason, we focused on males, and considered all the adult males of the community at that time ($N = 13$). Two of them disappeared between the start and the end of the study. Two additional individuals could not be tested. Oner was never leaving the main group on its own as he was the alpha male over the course of the study. The other was not well habituated and showed signs of nervousness when being followed. We were thus left with $N = 9$ individuals.

## Procedure

Figures 1A, 1B sums up the procedure. Individuals were distributed into two groups: the Social ($N = 5$) and Non-Social ($N = 4$) groups. Fields conditions prevented random subject allocation to the two groups. The individuals that were more likely to be found alone were

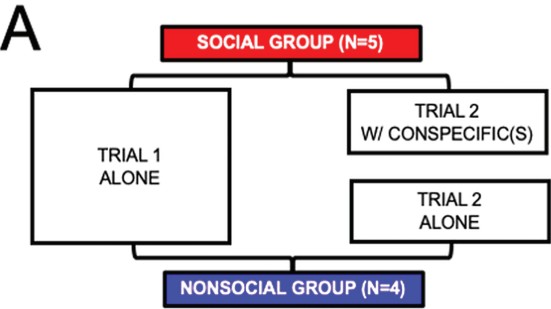

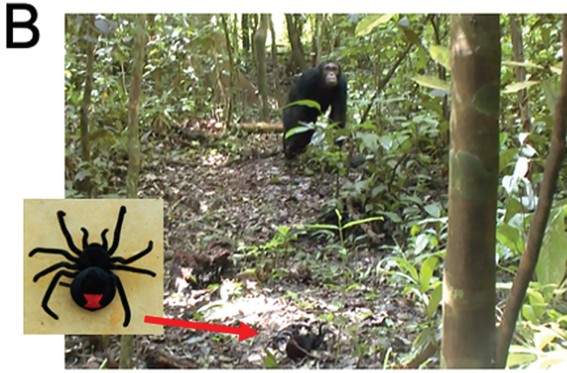

**Figure 1** (A) Summary of the procedure. Both Social ($N = 5$) and Non-Social ($N = 4$) groups underwent two trials. Trial 1 was alone for individuals of both groups. Trial 2 was with at least one conspecific for individuals of the social group, and alone for individuals of the Non-Social group. (B) Photographs of the spider (left image) and example of one individual (FK) inspecting the spider during a trial.

selected to be part of the Non-Social group. This introduces a potential confound which we will elaborate on in the Discussion section.

All individuals were tested twice. In the first trial, individuals were alone (*i.e.*, no other individual was present within a 30-m radius for at least 30 min and no calls from community members had been heard from within 100 m for at least 10 min). In the second trial, individuals from the Social group were tested with at least one member of their community present within a 30-m radius. Individuals from the Non-Social group were re-tested alone. The gap between the two trials was 10.57 days (+/− 15.30 SD) on average.

Each trial consisted in the presentation of a mechanical spider to the focal individual (as done in *Dezecache, Crockford & Zuberbühler (2019)*). The spider was set at around 7-m distance from the animal and operated for duration 18.45 s (+/− 8.35 SD). This model represents a species absent to the fauna of the Budongo Forest reserve, meaning it represents a stimulus worth communicating about even if not immediately dangerous. To avoid chimpanzees inferring that the decoy spider was a human artifact, we carefully avoided manually interacting with it when in the presence of our subjects. We ensured the target saw the spider first by placing it closest to them.

In a pilot experiment (conducted a month prior), we exposed four male chimpanzees ($N = 2$ from each group) to the spider and found that it elicited hoo alarm calls in all

subjects, thereby suggesting that it was perceived as a likely threat. Because of this pilot, it may be argued that those four chimpanzees had an increased exposure to the target stimulus. This is unlikely to represent a confounding issue, for two reasons: firstly, there was a month-long time interval between pilot and experiment; secondly, all individuals started with an 'alone' trial when the experiment began, and the statistical comparison is being made between the two experimental trials for each individual of both groups.

## Coding

As main alarm responses, we considered startle/alert hoos (*Crockford et al., 2012*; *Crockford, Gruber & Zuberbühler, 2018*; *Schel et al., 2013*) and alarm barks (*Crockford & Boesch, 2003*). We also examined the gazing behavior, since vigilance or gaze-alternation patterns are known to be modulated by communicative goals (*Crockford, Wittig & Zuberbühler, 2017*; *Dezecache, Crockford & Zuberbühler, 2019*; *Schel et al., 2013*). Specifically, since chimpanzees are adept gaze-followers (*Call, Hare & Tomasello, 1998*), we reasoned that participants could manipulate others' attentional states by maintaining gaze towards a relevant stimulus. This is consistent with marking behavior seen in chimpanzees upon encountering a snake (*Crockford, Wittig & Zuberbühler, 2017*).

We coded whether there was calling behavior (yes = 1; no = 0) by the subject from detection up to 30 s. We also coded from the videos using software BORIS (v8.5, *Friard & Gamba, 2016*) the amount of time the subject was looking (*i.e.*, gaze marking) at the spider from detection up to 30 s. In the two trials of the individual PS looking time measurement was interrupted after ~16.5 and ~25 s respectively because the individual moved location and was not in a position to see the spider anymore. Visual examination from the refuge location was done immediately after the trial to confirm this. To avoid disparities across individuals, we divided the looking time by the total possible looking time (usually 30 s, except for the two trials involving PS), and obtained the proportion of looking time within the possible looking time window for each individual and each trial.

Inter-rater reliability test based on $N = 17$ trials found consistency between GD and two naive observers on looking duration (Single Score Intraclass Correlation: ICC(C, 1) = 0.828; $F(16, 32) = 15.5$; $p < 0.001$). One trial was discarded for this test because the two naive observers did not pay attention to the event of interest (*i.e.*, the spider beginning movement). Raters were also consistent on how long the spider was activated for (ICC(C, 1) = 0.996; $F(16, 32) = 781$; $p < 0.001$). Note that, since alarm calls can be very soft in chimpanzees, they were coded live together by GD and SA and/or MM.

## Statistical analysis

Statistical analyses were conducted using R (v. 3.5.1; *R Core Team, 2020*) and RStudio (v. 1.1.456; *RStudio Team, 2020*). To test for a Group effect on call production and gaze marking in trial 2, given the call baseline obtained in trial 1, we ran general linear mixed models with a binomial distribution using function 'glmer' of package 'lme4' (v1.1.23; *Bates et al., 2007*). The dependent variable for calling was whether calling occurred or not and for gazing was whether the subject gazed or not in the 30 s following stimulus detection. The models included predictors 'Group', whether calling occurred or not at trial

1 (for the vocal warning model), proportion of time spent looking at the spider in trial 1 (for the gaze marking model), the delay between trials 1 and 2 (in days) ('Delay'), and the length of spider activation in the two trials (in seconds) ('Spider movement duration at trial 1' and 'Spider movement duration at trial 2'). Although similar studies controlled for startle responses (*e.g.*, *Crockford et al., 2012*) we decided not to control for this variable in our model owing to a small number of cases of startle responses (see Table S1). To investigate the effect of Group, we compared our full model with a null (intercept-only). Our statistical procedure was the same for testing a Group effect on gazing at the spider.

Before models were run, we transformed (log-transformation followed by z-scoring) predictor 'Delay' because its distribution suffered from right-skewedness. We also z-scored predictors 'Spider movement duration at trial 1' and 'Spider movement duration at trial 2'.

## RESULTS

### Vocal warning

During the first exposure to the spider model, 5 of 9 alone subjects produced alarm calls (three of four non-Social group and two of five Social group subjects). No vocal or behavioral responses were heard from conspecifics, confirming they were alone. During the second exposure, the proportion of calling individuals in the Non-Social group decreased from 3/4 to 1/4 subjects, whereas in the Social group, calling increased from 2/5 to 3/5 subjects (Fig. 2A). All calls were alarm hoos (*Crockford et al., 2012*; *Crockford, Wittig & Zuberbühler, 2015*; *Crockford, Gruber & Zuberbühler, 2018*), in addition to one pant-hoot call (*Fedurek, Zuberbühler & Dahl, 2016*) produced by individual KT (Social group) on trial 2. Although the patterns observed are anecdotally consistent with our predictions (see Fig. 2A), our full model was not statistically better than the null model (Type II Wald chi-square tests: X(4) = −2.65; *p* = 0.61). The full model summary is shown below in Table 1 and raw data are shown in Table S2.

### Gaze marking

During the first exposure, the mean looking duration at the spider was 21.07 +/− 7.86 s (Mean +/− SD), with no obvious differences between the subjects assigned to Social and Non-Social groups (Social = 22.2 +/− 7.81 s *vs.* Non-Social = 21.4 +/− 9.17; Means +/− SDs). During the second exposure, looking duration decreased overall to 15.98 +/− 6.71 s (Means +/− SDs), especially in the Non-Social group (Social = 18.8 +/− 3.38 s *vs.* Non-Social = 9.75 +/− 2.43 s; Means +/− SDs). Correspondingly, our full model was significantly better than the null model (Type II Wald chi-square tests: X(4) = 29.37; *p* = < 0.001). The full model summary is shown below in Table 2 and raw data are shown in Table S3.

*Post-hoc* tests showed that, at first exposure, looking time did not differ between groups (t ratio test: t-ratio = 0.16; *p* = 0.87) but a significant difference emerged at the second exposure, with looking time being higher for the Social than Non-Social group (t ratio = 2.22; *p* = 0.046, d = 2.80), a result which is significant with a two-tailed test without correction for multiple comparisons (in that case, two comparison). All other model
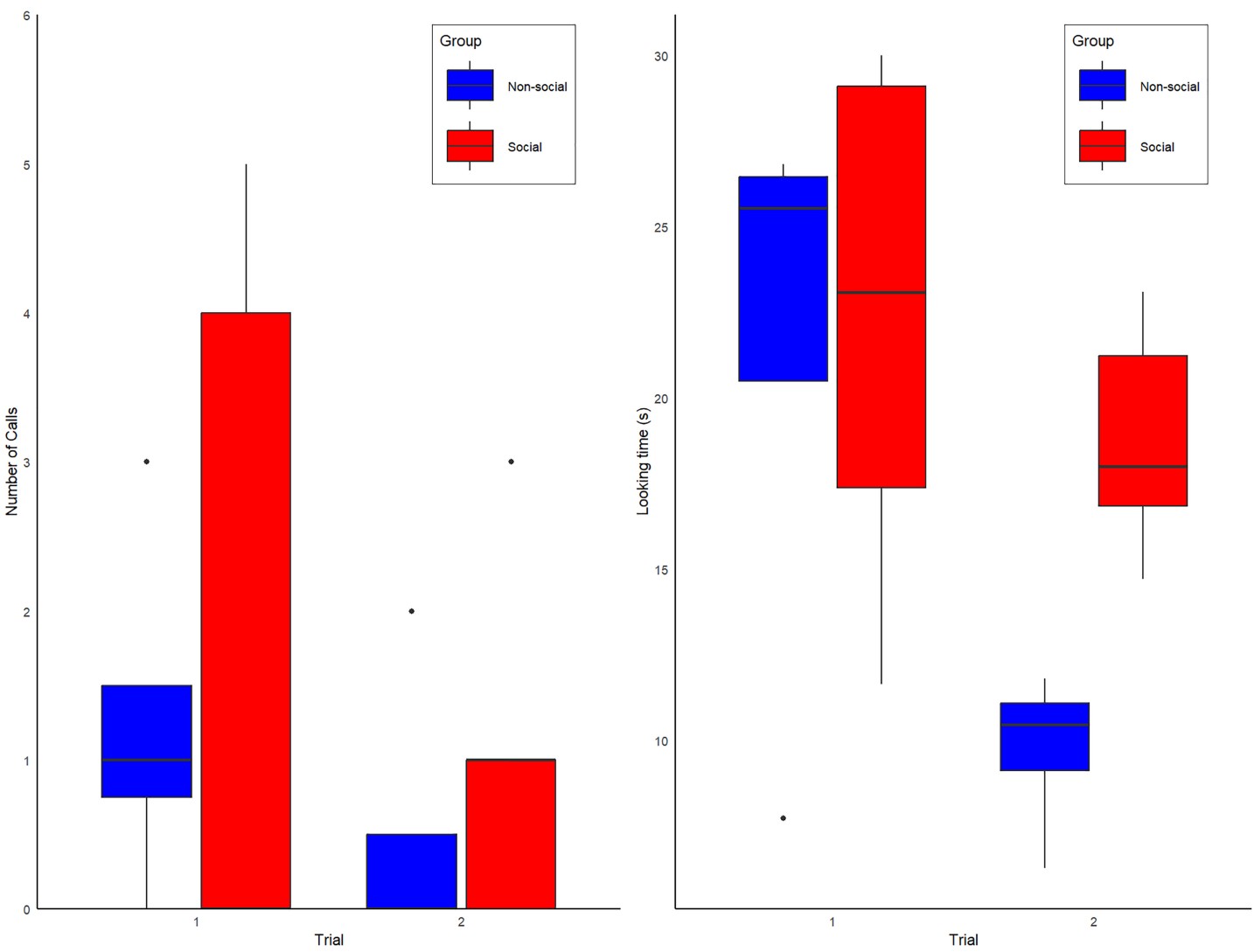

**Figure 2** Box plot graphs showing the (A) number of calls observed for the Social (red) and Non-social (blue) groups for trials 1 and 2. (B) Overall amount looking time spent looking at the spider for individuals of the Social (red labels) and Non-Social (blue labels) groups.

factors (Group, Delay between trial 1 and 2, duration of spider activation) were non-significant (all ps > 0.1). Looking time data are shown in Fig. 2B.

## DISCUSSION

We asked whether the presence of an uninformed audience increases the motivation in wild chimpanzees to communicate about an unfamiliar and potentially dangerous target, which the signaller had already been exposed to. We adopted an experimental approach wherein we presented nine subjects with a potential threat twice, first in private without any audience and then a second time either in private again or with a naïve audience within visual range. We measured two indicators of warning: alarm calling and gaze marking. We found that subjects decreased the production of warning behavior if encountering the

**Table 1 Vocal warning full model summary.**

|                    | Estimate | Standard error | Z value | p-value |
|--------------------|----------|----------------|---------|---------|
| Group              | 2.01     | 1.83           | 1.09    | 0.27    |
| Spider movement    | 0.14     | 0.97           | 0.15    | 0.87    |
| Delay              | 1.24     | 1.23           | 1.01    | 0.31    |
| Alarm call at trial 1 | −0.95 | 1.94           | −0.49   | 0.62    |

**Table 2 Gaze marking full model summary.**

|                    | Estimate | Standard error | Z value | p-value |
|--------------------|----------|----------------|---------|---------|
| Group              | 1.03     | 0.27           | 3.75    | <0.001  |
| Spider movement    | 0.01     | 0.02           | 0.90    | 0.36    |
| Delay              | −0.30    | 0.18           | −1.61   | 0.10    |
| Proportion of looking in trial 1 | −0.66 | 0.71 | −0.92 | 0.35  |

threat twice in private (Non-Social group). By contrast, if subjects were in the presence of uninformed others when encountering the threat a second time (Social group), they produced comparably higher rates of communicative behavior.

In the Non-Social group, alarm calling decreased from three to four to one of four alarm calling subjects, whereas in the private-social condition it increased from two of five to three of five subjects. Although this change was not statistically significant, the observed changes across conditions were consistent with the hypothesis that while calls may have initially been arousal driven, since most individuals showed alarm calls on first exposure, the later production of the target behaviors was modulated by the presence of uninformed others. This interpretation aligns with the finding of *Girard-Buttoz et al. (2020)* who found chimpanzees were more likely to produce alarm calls in response to predator models when they had not yet heard an alarm call from a conspecific. Given that the animals tested in the pilot showed consistent alarm behavior in response to the decoy spider, it is unlikely that the lack of group differences in warning behavior reflected a categorization of the target stimulus as irrelevant or non-threatening. Instead, we suggest that alarm calling may have required additional experience with the stimulus to be communicated about.

Indeed, in another study using a similar approach, older chimpanzee individuals were more likely to produce alarm calls in response to the stimulus while gazing behaviors (measured through gaze alternations) were common at all ages (*Dezecache, Crockford & Zuberbühler, 2019*). These findings suggest that, in contrast to several monkey species wherein juveniles but not adults show alarm responses to novel predators (*Leon et al., 2022*; *Mohr et al., 2023*), reliable alarm calling behavior in chimpanzees requires more extensive prior experience of danger, while gazing behavior may not. Moreover, we believe our findings point towards the involvement of other cognitive mechanisms such as the formation of long-term memories on a single-trial basis, since effects were observed at least 10 days after a single experience of the novel stimulus.

Changes in looking behavior were instead significant, being higher in the second trial in the social compared to non-social condition. Although gaze has non-communicative functions such as visual perception, thereby explaining generally high amounts of gazing in trial 1 across groups, gaze is indeed known to play an important role in primate communication. In a study of young chimpanzees in response to a novel threatening model, gaze alternations were consistently observed to co-occur during alarm calling (*Dezecache, Crockford & Zuberbühler, 2019*). Similar patterns have been observed in response to more familiar predators and are commonly interpreted as fulfilling a 'marking' function that allows conspecifics to locate danger (*Crockford, Wittig & Zuberbühler, 2017*; *Schel et al., 2013*). We could not measure gaze alternations in this study because in the non-social condition there would be no social partner to gaze towards, creating a bias between conditions. It nonetheless seems plausible that the gaze behavior observed in the social condition also fulfilled a communicative function, consistent with a motivation to inform uninformed conspecifics by drawing their attention to potential threats *via* sustained looking to the visual referent.

This said, we concede that the evidence presented here is provisional and open to competing interpretations: for instance, it remains possible that individuals produced more looking behavior towards the decoy spider in the presence of conspecifics because of the increased arousal of the focal individual due to the mere presence of others or in light of the potential threat that the spider may have represented for other group members. These alternative explanations cannot be presently ruled out insofar as the first encounter with the stimulus was always asocial. To test for such confounds, further experimental work could present individuals with the stimulus on first encounter in the presence of conspecifics. If individuals are motivated to inform others, we would expect communicative behaviors to not increase in the second trial if the signallers have evidence that the potential audience was already knowledgeable about the about the threat.

One important methodological limitation in the present study is the lack of randomization in group assignment: individuals that we encountered alone were opportunistically included in the Non-Social group. According to the STRANGE framework (*Webster & Rutz, 2020*), this may give rise to a self-selecting bias: perhaps individuals in the Non-Social group were more likely to be found alone because they are less inclined to socialize with others. This is a crucial confounding factor, insofar as individuals who are disposed to be less social may also be less motivated to inform others. Buttressing this possibility, it has been recently shown that more sociable chimpanzees are more persistent in their gestural communication (*Roberts & Roberts, 2019*). Tempering the worry that self-selection biases might explain the observed pattern, individuals in each group produced consistent and comparable alarm responses to the stimulus during the pilot phase. For this reason, self-selecting bias between experimental conditions is unlikely to explain the observed patterns.

## CONCLUSIONS

To conclude, in the present study we aimed to examine whether chimpanzee produced alarm calls and gaze marking to inform others when already familiar, and thus plausibly no

longer aroused, by the relevant stimulus. While further experimental work remains to be done in order to provide stronger evidence of this motivation in chimpanzees, our research represents an important first step, as it introduces an original and potentially fruitful field research protocol and provides further support to the claim that chimpanzee communication is driven by the motivation to inform others.

## ACKNOWLEDGEMENTS

Our gratitude goes to Geoffrey Muhanguzi and Caroline Asiimwe.

### Funding

The Fyssen Foundation fellowship was awarded to Guillaume Dezecache and funding through European Union's Seventh Framework Programme for research under grant agreement no 283871. The NCCR Evolving Language, Swiss National Science Foundation (#51NF40_180888) supported the APC. The funders had no role in study design, data collection and analysis, decision to publish, or preparation of the manuscript.

### Grant Disclosures

The following grant information was disclosed by the authors:
European Union's Seventh Framework Programme: 283871.
NCCR Evolving Language, Swiss National Science Foundation: #51NF40_180888.

### Competing Interests

The authors declare that they have no competing interests.

### Author Contributions

- Derry Taylor analyzed the data, prepared figures and/or tables, authored or reviewed drafts of the article, and approved the final draft.
- Sam Adue performed the experiments, authored or reviewed drafts of the article, and approved the final draft.
- Monday M'Botella performed the experiments, authored or reviewed drafts of the article, and approved the final draft.
- Denis Tatone conceived and designed the experiments, authored or reviewed drafts of the article, and approved the final draft.
- Marina Davila-Ross conceived and designed the experiments, authored or reviewed drafts of the article, and approved the final draft.
- Klaus Zuberbühler conceived and designed the experiments, authored or reviewed drafts of the article, and approved the final draft.
- Guillaume Dezecache conceived and designed the experiments, performed the experiments, analyzed the data, prepared figures and/or tables, authored or reviewed drafts of the article, and approved the final draft.

## Animal Ethics

The following information was supplied relating to ethical approvals (*i.e.*, approving body and any reference numbers):

Uganda National Council for Science and Technology.

## Field Study Permissions

The following information was supplied relating to field study approvals (*i.e.*, approving body and any reference numbers):

Uganda Wildlife Authority.

## Data Availability

The raw data and r script for reproducing our analyses of the data are available in the Supplemental Files.

## Supplemental Information

Supplemental information for this article can be found online at http://dx.doi.org/10.7717/peerj.18498#supplemental-information.

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
