# Peer review of "The motivation to inform others: a field experiment with wild chimpanzees"

_PeerJ, doi:10.7717/peerj.18498_

## Round 0.1 · original submission · Major Revisions

The manuscript examines alarm calling behavior in chimpanzees and bonobos to determine if it is driven by socio-cognitive processes or merely by arousal. Reviewers identified several key issues: the statistical analysis was over-parameterized with a small sample size, the introduction needs to better articulate the study's novelty, and methodological clarifications are required, such as the criteria for defining adult males, the novelty of the spider model, and the analysis of looking time. The abstract should accurately present results, and the discussion should consider both emotional and cognitive drivers of alarm calls. Additionally, the manuscript should emphasize the novelty of using a completely new stimulus. Addressing these points will enhance the study's clarity and rigor. Major revisions are needed.

·

Basic reporting

Maël Leroux

The article proposed by Taylor et al is very informative and investigate further the cognitive capacities underlying intentional communication and theory of mind in our closest living relatives: chimpanzees. The author show experimentally that chimpanzees tailor their communicative behaviour according the state of knowledge of their conspecifics. The experimenal design is well thought out and allow robust conclusions. The paper is very well written and the significance of the findings is extremly relevant to the current research environment and help move the fields of comparative cognition, comparative language science, and animal linguistics forward. I support the pupblication of the current paper in Peer J.

I have a few suggestions that I feel could help the authors improve further the paper.
In general, I think the logic behind the study could be more explicit. It is not clear to me if by presenting the spider twice, you expect the subject to habituate to it, i.e. you predict that the first interaction will be sufficient to the subject to integrate this novel item as a low-urgency threat and hence respond less to it on the second presentation when alone. In that case, would 10 days interval between presentation 1 & 2 allow for such habituation? I wonder what this would tell us about the cognitive capacities of chimps. Would it mean that whilst the subject understand the spider is a low urgency threat, they still alert conspecific when uninformed individuals come by, i.e. do they intend to communicate the presence of a threat, or would they intend to prevent an unaware individual to be startled when seeing the spider? Although I understand it’s difficult to confidently conclude on any of that, these are questions that I had when reading the paper and I think would be worth discussing for the reader. This study would not only be relevant for intentional communication, but could also have interesting implications for theory of mind in chimps.
Finally I would review the Schel et al 2013 paper (plos one) that could be put to a better, more extensive use here – I think the present study explicitely tested what Schel et al. may have implied.


L15: Behaviours, UK english needs to be consistently applied throughout the MS (or US, but consistent).
L50: not a native speaker, but it seems to me a word is missing here.
L57-68: I feel a key study is missing here, which answers at least part – if not all – your concerns here: Schel et al., 2013. They present snake models to chimpanzees in three conditions: alone, when the chimpanzee is in the back of the party and when a chimapnzee is in the front. Here the exposure was repeated, and hence you could argue chimpazees do indeed respond to the presence of ignorant group members rather than the threat alone. I don’t think this takes anything out of your study though, as Schel’s et al. did not explicit test this, and the experimental design you present here test this hypothesis much more explicitely.

Experimental design

no comment

Validity of the findings

no comment

·

Basic reporting

In this study the authors draw on previous field experiments on chimpanzees and bonobos to assess whether alarm calling behavior in the presence of a potential danger is truly driven by socio-cognitive processes, as previously argued, or simply a by-product of caller arousal. To do so, the authors designed a novel experimental paradigm whereby they prime study subjects with a model spider first all in a situation where they are socially ‘alone’ and second either alone (non-social group) or in the presence of conspecific (social condition). Individuals in the social condition on the second trial are thus potentially less aroused by the spider model and the production of alarm calls is considered as motivation to inform others rather than emotionally driven. The authors also considered how long the study subject looked at the snake when alone or in the presence of others as further indication of the motivation to inform others visually. The authors found calling behavior consistent with the predictions of the chimpanzee call to inform others since less individuals called the second trial when alone twice whereas the calling remained similar and even slightly increased between alone and social condition. These results were however not statistically significant, due to the limited sample size in this study. This lack of significant could also be driven by an over-parametrization of the statistical model with 4 predictors for only 18 data points. I would recommend to the authors to use only 2 predictors (social/non-social group and trial 1 or 2) and describe the effect of the other two in the text.
In a second set of analyses, which are not clearly explained (what error structure did you use to analyze proportion of looking time) the authors found significant increased looking time towards the snake when in the presence of conspecifics.
Overall the manuscript is well written and clear. The introduction should reflect more accurately what has been analysed in previous study and report more clearly on why this study is truly novel. The analyses for the second model should be clarified. In the discussion, the authors should expand on the possibility that first alarm calls upon discovery of a potentially threatening objects are likely emotionally driven since they found a large number of males calling in the trial 1 when they were technically alone. Second they should mention that nevertheless the rest of the calling behavior is likely not only emotional and is possibly triggered by the motivation to call others.

Abstract

Line 17-18: This claim is a bit too strong since previous studies already accounted for the arousal of the subject by determining whether they startled or not upon discovering the snake (Crockford et al. 2012; Girard-Buttoz et al. 2020). Furthermore, the fact to present repeatedly the same stimulus (a snake in other studies) is not new since some presentation were repeated on the same location with the same subject in Girard-Buttoz et al. 2020. The authors should be more explicit about what is truly new about their study or present it rather like an expansion/confirmation than a complete novelty. The later does not impact the value of the study.
Line 27: Gaze-marking is not self-explanatory, maybe use another term or define it briefly in the abstract? Also you present results as fact whereas for the calling behavior your statistical produce non-significant results. You should rephrase this presentation in the abstract to reflect this fact.



Introduction
The introduction is overall very clear and well written. It lacks however certain level of details on the two main studies cited Crockford et al. 2012 and in Girard-Buttoz et al. 2020. In particular, it is important to mention that both studies di not just ignore the possibility that the calling behavior was driven by emotional state and surprise of the individual discovering the snake. They did account for that to some extent by including ‘startled’ Y/N in their statistical analyses and also by sometime presenting the snake on the same location several times so that certain informers would already be familiar with the stimuli. The authors have an argument to claim their study is novel, namely that they explicitly control for exposure to stimuli and possibly that the stimuli is more novel (more details bellow) than a snake model mimicking a species they truly encounter. Another point is that informing ignorant pattern has also been found in bonobos and to make the introduction less chimpanzee focused this should be explicitly stated throughout (more details below).
Line 41: in Crockford et al. 2012 and in Girard-Buttoz et al. 2020 the authors specifically controlled for the number of time each individual have been exposed to a snake model in the study. Please mention that point here, as well as the fact that arousal was controlled for considering whether the individuals startled or not upon sein the snake. Explicitly state then why you think that is not sufficient to account for the varying degree of experience with the subject.
Line 42: remove the ‘the’ in the sentence.
Line 60: Mention that this finding was replicated in bonobos so that it is not a chimpanzee-only pattern (Girard-Buttoz et al. 2020).
Line 60: This statement is not correct, as mentioned above in Girard-Buttoz et al. 2020 the model was sometimes repeatedly presented to the same individual on the same location on consecutive days.
Line 66-68: This seems unlikely since in both Crockford et al. 2012 and in Girard-Buttoz et al. 2020 the authors controlled in their analyses for the ‘surprise’ by including the factor ‘individual startled upon discovering the snake or not’. Maybe mention that specifically and explain why your approach is novel in itself. Maybe emphasize the fact that this was just a control whereas previous exposure to predator was not manipulated in a controlled and balanced design. The novelty in your study as far as I can tell is that you present a potential predator that is completely novel whereas the snake try to mimic a natural predator they encounter. In this sense it is hard for snake experiments to control truly for individual familiarity with snakes in general and in particular Gaboon and rhinoceros vipers since that would require constant monitoring of their encounters with real snakes, which is impossible.
Line 73-77: Again, please add the bonobos into the picture to make it more general. (by adding Girard-Buttoz et al. 2020 that also found that bonobos stop calling when the others are out of dangers because they have seen and identify the location of the snake).
Line 95: It would be important to state whether the spider used in the experiment aims at mimicking a naturally occurring spider species or was a completely new stimulus.

Methods
The methods are overall clear. I am not sure why the authors did not control for parameters such as ‘startled’ Y/N or at least report on it in te results. I understand that with 18 trials, it is hard to control for everything but it would be useful to report on the number of individuals who startled when exposed to the spider the first and the second time. The second model with proportion is either not correctly analyzed or the authors do not specify enough how the error distribution differed from the first binomial model with 0 and 1 responses. Proportion should be analyzed using beta model.

Line 129: Say how many individuals are in the community vs. how many are fully habituated to human observers.
Line 134: What is your criteria to define adult males?
Line 158: What do you mean by ‘operated’? Was the spider moving? This is important information. Also because it contrasts with some snake experiments where the presented model was static.
Line 159: The information about the novelty of the subject to the chimpanzee and its absence from the natural fauna should be mentioned and used as an argument in the introduction already to explain the novelty of the study and the difference to precedent approaches (see comments above).
Line 184: why 30 seconds?
Line 183-193: Did the authors look specifically at gaze alternation between the spider and the individual in the audience (in particular the ones approaching the spider)? That would be important to mention and also state if and how often this happened. I am aware as you state in the discussion that you cannot measure this parameter in the ‘alone’ condition but it would be interesting to report its occurrence for social condition, even without statistical analyses.

I am not sure why they analyzed the looking time only for 30 seconds. It seems that the looking time is relevant in the presence of others than can arrive at the spider at different point in time. Or were the party always so cohesive so that all individuals around the focal subject could be in proximity and see the danger within 30 seconds?

Line 198: can you clarify what you mean here? I do not understand this sentence sorry. What is the ‘accurate event of interest’?

Line 200-201: How were the alarm calls coded ‘live’? did the observers make vocal commentaries directly into the microphone?
Line 207: Did you use a binomial model? If so please specify that is easier to follow for non-specialist readers.
Line 208: Is there a reason why the authors did not control for whether the subject startled upon discovering the spider? This would be important to control for and also would provide important information to see if the spider is perceived as less threatening when presented in a different location but to the same subject. In fact, that is a big assumption of the current study and it would be good to quantify if this is true or not. Of course what is missing in the procedure is individuals that were presented the spider for the first time in a group situation. That also raises the question how did the authors manage to have the focal subject see the spider first in the social setup? What happened if another individual saw the spider first? If so that would also answer the question above.
Is there a reason why you did not also analyze how many calls the focal subject produced on top of whether they called or not? I would expect the difference between social and non-social to be even more important.

Line 208-212: This model appears over parameterized, especially when using frequentist approach having more than 2 predictors in a model with less than 20 data points is not recommended. The authors could however report on the effects of delay and call first trial Y/N numerically in the result text.

Line 214: While the control variables were the same, you cannot analyze proportion using binomial model (which is what you did in the first model I assume). Proportions are bound to 0 and 1 but are not either 0 or 1, and should be analyzed using a beta error distribution.


Results

Line 223: It is surprising that hoos were produced when individuals where alone. Where the hoos somehow louder and could they have been heard by individuals not considered ‘together’ with the focal subject?

Line 228: I don’t think the identity of the individual is relevant here especially since no individual information on the study subject is provided such as age or rank.

Line 233: Please provide result of the full-null model comparisons for this analysis as well.


Discussion

The authors should discuss the possibility that alarm call production is at first emotional since many individuals produced alarm calls in the alone condition but that the reiteration of calling upon being less aroused or familiar with the stimulus may be more cognitively driven and controlled. Discuss how this pattern also possibly matches what has been found in previous studies. The authors should also acknowledge that the proposition that chimpanzees (and bonobo) alarm calling behavior is driven by motivation to inform others (as exemplified by the chimpanzee bonobo difference Girard-Buttoz et al. 2020).

Line 297: remove the ‘the’ after ‘increased’.
Line 302: remove the last ‘to’ in the line.

Experimental design

see report above

Validity of the findings

see report above

---

## Round 0.2 · Minor Revisions

Please make the final revisions to your paper based on the latest minor comments, and we will proceed with its acceptance in PeerJ. Best regards,

·

Basic reporting

I thank the authors for including all my points.
I think the paper has improved as a result.
I do not have any concern remaining.
Find minor typos corrections in the attachement (although reading my previous review, we may need to ask reviewers to review my reviews...)
Thanks for a great paper!
Maël

Experimental design

NA

Validity of the findings

NA

Additional comments

NA

·

Basic reporting

I appreciate the effort the authors put into revising their manuscript that has now improved both in clarity and accuracy. I still since yet that some effort needs to be put to clarify even further the rational of the study and especially the gain to controlling for previous exposure in order to ascertain the effect of knowledge status on information transfer behavior. On this note, it is not clear to me how the current protocol addresses audience knowledge since there is only 1 social condition with uninformed individuals being compared to an asocial condition. The authors discuss this possibility in the discussion but should be more upfront about this drawback already when introducing this study. I provide more detailed comments below.


Please check for spacing throughout, there are numerous places with large spaces between words.

Line 36: Specify that you are talking abut chimpanzees and bonobos.
Line 47: remove the “the” after “account for” .
Line 48: What you say here is true but it is not straightforward how varying experience with snakes in their environment could account for them calling when ignorant are present in the audience and stopping calling when all individuals are informed. Maybe you want to clarify the link here and the point you are trying to make since it is quite crucial to the theoretical rational of the paper.

Line 50-57: Somewhat it feels that the introduction might be constructed backward. This paragraph is much more general than the first paragraph and the authors might consider putting it first.

Line 66: This study di not only study bonobos but expanded findings from Crockford 2012 to another chimpanzee population, enhancing the generalizability of the phenomena to different chimpanzee populations. This is a point the authors should mention.

Line 75: Again, it is not clear here why lack of control for previous exposure would affect the result regarding audience knowledge. Also please note that in previous study infants were excluded and given that chimpanzees from the same group range in the same territory it is unlikely that the rate of encounter with natural snake varies largely between individuals, at least probably not enough to create strongly different level of experience between them. The authors should clarify there rational here.

Line 79-81: I am not sure I understand the argument. Detector have to be the first to see the snake and therefore are by definition not informed. Informed individuals do return to the snake too, especially to share experience with others (e.g., new arrivers in the party).

Line 105: While previous study does not control as you say exposure to natural snakes, they do however proceed to several exposures of the same snake model to the same individuals either on the same day or even on consecutive days. The fact that individual who have already been exposed before do call anyway when ignorant are present even on the 3rd or 4t day of exposure indicates that this is not about surprise.

Line 112: I do understand that surprise was not controlled experimentally but if I am not wrong you also don’t present several different novel stimuli to the same individuals. In that sense you may want to explicitly state how your approach differs from controlling for surprise previously conducted.

Line 120-127: If I understand correctly you compare responses between first exposure where they are alone to second exposure either with others or alone again. At the end you have 2 social conditions, either alone or with others and I am not sure I understand how you disentangle the fact that they call to inform ignorant others or that they call just because there is an audience (which is what has been shown in many animals and is general social audience effect without need to process cognitively knowledges status). The later process that is much less cognitively demanding (I don’t call alone I call with an audience) would lead to the exact same results as if they were truly informing ignorant others. Or am I missing something? My point is that emotions can also drive audience effect without needing to be intentional (the caller is aroused by the presence of others and therefore calls, independent of the previous exposure with the stimuli). Maybe the authors should make clear here how the two possibilities can be disentangled.


Results

Line 251: The fact that 5 out of 9 subjects alarm called here suggests that an audience could hear the calls even if they were alone by your definition. It would be important to mention if following the calls other chimpanzees from other parties approached the caller and join them. If so that suggests some limitation in the definition of the alone condition that should be discussed.
Line 291: One should be careful about the interpretation of the gazing behavior here. The fact that they gaze longer the first time they encounter the spider suggests that the primary aim of gazing is not to inform others (since they were alone) but to gather information about the potential threat. This aspect should be discussed.

Experimental design

see above

Validity of the findings

see above

Additional comments

see above

---

## Round 0.3 · accepted · Accept

The authors have addressed all of the reviewers' comments. This manuscript is ready for publication.